# Multi-Sensors Enabled Dynamic Monitoring and Quality Assessment System (DMQAS) of Sweet Cherry in Express Logistics

**DOI:** 10.3390/foods9050602

**Published:** 2020-05-08

**Authors:** Xiaoshuan Zhang, Xuepei Wang, Shaohua Xing, Yunfei Ma, Xiang Wang

**Affiliations:** 1Beijing Laboratory of Food Quality and Safety, College of Engineering, China Agricultural University, Beijing 100083, China; zhxshuan@cau.edu.cn (X.Z.); sy20193071259@cau.edu.cn (X.W.); SYUleiwang@cau.edu.cn (Y.M.); 2College of Food Engineering, Ludong University, Yantai 264025, China; xshjob@163.com

**Keywords:** sweet cherries, express logistics, dynamic monitoring, quality assessment, multi-sensors, freshness prediction

## Abstract

The market demand for fresh sweet cherries in China has experienced continuous growth due to its rich nutritional value and unique taste. Nonetheless, the characteristics of fruits, transportation conditions and uneven distribution pose a huge obstacle in keeping high quality, especially in express logistics. This paper proposes dynamic monitoring and quality assessment system (DMQAS) to reduce the quality loss of sweet cherries in express logistics. The DMQAS was tested and evaluated in three typical express logistics scenarios with “Meizao” sweet cherries. The results showed that DMQAS could monitor the changes of critical micro-environmental parameters (temperature, relative humidity, O_2_, CO_2_ and C_2_H_4_) during the express logistics, and the freshness prediction model showed high accuracy (the relative error was controlled within 10%). The proposed DMQAS could provide complete and accurate microenvironment data and can be used to further improve the quality and safety management of sweet cherries during express logistics.

## 1. Introduction

Sweet cherries *(Prunus avium)* are rich in anthocyanins, quercetin, hydroxycinnamate, fiber, vitamin C, carotenoids and melatonin, which have potential health benefits for cancer, cardiovascular or inflammatory diseases, higher nutritional value attracts increasing consumers [1,2]. In addition, compared with other fruits (apple, peach, and pear), the earlier ripening within the soft season, unique taste and high economic value have also promoted sweet cherries taking a growing share of the fruit industry [3,4,5]. The cultivation of sweet cherries has become one of the typical flourishing high-efficiency industries in China [6,7]. According to statistics from the UN Food and Agriculture Organization, the area planted with sweet cherry in China had quadrupled from 2000, and the yield in 2017 reached about 400,000 tons. It is predicted that the sweet cherries planting area and output will reach about 200,000 hm^2^ and 1.2 million tons respectively in China by 2025 [8].

Although the sweet cherries industry is very popular in China, several factors are limiting its further development. Firstly, sweet cherries have high water content, short shelf life, and are extremely perishable and deteriorated with a limited shelf life of 7–10 days after picking [9,10]. Secondly, sweet cherries are a non-respiration climacteric fruits, which means that ripening does not continue the following harvest and therefore cannot be harvested until fully ripened. The typical harvest period of sweet cherries is typically from May to July when the outdoor temperature can reach 35–40 °C Thirdly, the planting area is mainly distributed in Shandong province (82,000 hectares) and Liaoning province (2.9 million hectares) [11], which means that the distribution of its planting area is extremely uneven in China. Therefore, its property, picking condition, and uneven planting distribution pose a huge obstacle to the market supply period and off-site sales of sweet cherries.

As the development of the Internet economy and e-commerce platforms such as Alibaba, JD, and Pinduoduo, in 2018, more than 1.3 billion tons of fresh agricultural products were sold online in china [12]. The “Producer-E-commerce-Consumer” sale model has occupied increasing shares in fresh sweet cherry sales models [13]. The realization of this sale model mainly depends on express logistics and cold chain logistics. The express logistics refers to the delivery activity which is accomplished promptly within the promised time frame according to the definition of the Postal Law of the People’s Republic of China [14]. The definition of the cold chain is to maintain the quality of fresh food and frozen food. The cold chain is a physical process that dominates the supply chain logistics of certain processed foods, which is always at a low temperature and has a logistics network with special equipment [15,16].

The sweet cherries are delivered to the consumers through express logistics, which effectively alleviates the problems of uneven planting distribution and off-site sales. However, owing to lower profits, it is unrealistic to realize the full cold chain transportation for most sweet cherries [17]. In addition, compared with cold chain logistics, express logistics has the following characteristics: the smaller size or weight (<30 kg), high timeliness, integrated, fragmented by time and space [14,18]. While the transportation mode of cold chain logistics is relatively fixed (including equipment, operating standards, transportation routes and duration, etc.). The transportation environment of fresh foods can be controlled well in cold chain logistics under the help of smart refrigeration and other equipment in cold chain trucks [19]. That means it is difficult to use traditional refrigeration equipment or wireless sensor network equipment widely used in cold chain logistics for express logistics.

Express companies often focus on timeliness and efficiency, while ignoring the importance of quality monitoring and management during the express logistics. If the postharvest quality management of fresh in-transit sweet cherry cannot be controlled well, a huge quality loss would be caused, seriously reaching a 20%–30% rot rate and furthering causing economic loss [10,11]. The decline in quality often results in complaints from consumers, and demand for compensation. These problems have brought great economic losses to farmers, consumers, and express companies. Although some express companies have gradually begun to take some measures to guarantee the quality of products, such as the application of active packaging methods and novel packaging materials before shipping, or priority delivery service for perishable products after shipping [9,20,21,22,23]. However, there is little attention to that process of express logistics. Besides, there are still some challenges for other technologies for solving the quality loss of sweet cherries in express logistics. For example, ZigBee technology relies on network and gateway communication, which is easy to cause data loss and measurement failure. The additional communication function brings certain challenges to power consumption and battery life [24,25]. As for the radio frequency identification (RFID) and near field communication (NFC) technologies, they are generally integrated into the hybrid monitoring system (as the smart tags, etc.), it is difficult for them to achieve multi-parameter real-time monitoring independently [26,27]. Our team carried out investigations on reducing the quality loss of products from the perspective of their transportation microenvironments, such as blueberry [28], Matsutake mushrooms [29], and North American holly [30]. However, research on the quality monitoring of pouch-packed sweet cherry in express logistics has not yet been reported.

Sensory evaluation was used to analyze the quality of food [31], while the method based on multi-sensors for dynamic monitoring quality analysis is becoming increasingly popular, because of its high precision, accuracy, reliability and multi-aspect. Multi-sensors technology has been already applied in the cold chain transportation of fresh agricultural products such as peach [32], pear [33], fish [34,35], shellfish [36], and got a good effect on quality management.

This study aims to reduce the quality loss of fresh sweet cherries in the express logistics. To accomplish such goal, a dynamic monitoring and quality assessment system (DMQAS) based on multi-sensors technology was proposed, thereby monitoring the key parameters of the microenvironment (including temperature, relative humidity, O_2_, CO_2_, and C_2_H_4_ content) during express logistics. Besides, a freshness prediction model of sweet cherries was established based on back propagation (BP) neural network. Combined with quality parameters (including hardness, soluble solid content (SSC), pH, and chromatic aberration), this system could provide a theoretical reference for the timely decision-making and management of express logistics processes by freshness prediction, which could reduce the quality loss in the express logistics of sweet cherries.

## 2. Materials and Method

### 2.1. Architecture Design and Implementation of the DMQAS

This section discusses the architecture design and implementation of the DMQAS, which consists of three components: hardware design, software design, and construction of freshness prediction model. The general framework of the DMQAS is shown in Figure 1.

In the express logistics of sweet cherries from the growing area to the consumer, the sensor nodes take data from the storage microenvironment at certain time intervals, including temperature, relative humidity, oxygen, carbon dioxide and ethylene content. Through the IO interface, the key parameter data of microenvironments are transmitted to the monitoring platform for analysis and furthering processing. After processing the data of key parameters, it is transferred to the application stage, including data management, data maintenance, data update. At the same time, the freshness prediction model retrieves the gas data from the corresponding database, conducts coupling modeling to predict the freshness of sweet cherries, and the predicted results are also stored in the database, which is convenient for users to view and invoke.

#### 2.1.1. Hardware Design

The monitoring platform was based on the STC12C5A60S2 chip (shown in Figure 2), which is mainly responsible for the accurate monitoring of the system, including the transmission of acquisition/sleep command, receiving and processing of data, and the control of operating frequency. The main control platform controls the sensor acquisition hardware to collect environmental parameters every 27 s, which was a trade-off between accuracy and power consumption. The monitoring terminal was powered by a portable lithium battery with a rated capacity of 30AH, an operating temperature of −20–60 °C and an operating relative humidity of 0–90%, which satisfies the actual environment of sweet cherry transportation. The power of O_2_, CO_2_ and C_2_H_4_ sensors are about 0.075 W, 0.0495 W, and 0.0495 W, respectively. Considering the power of the microcontroller (about 0.18 W) and power loss, the lithium battery can guarantee that the monitoring terminal could work for enough time (at least 180 h), which can meet most of the current express transportation time demands.

The sensing acquisition hardware is responsible for real-time monitoring of temperature, relative humidity, O_2_, CO_2_ and C_2_H_4_ in the sweet cherry transportation microenvironment, and converts the analog signals into digital signals for easy interaction with the control platform. According to actual investigation and review of literature, the gas composition involved in the sweet cherry transportation process is complicated, but the three gases of O_2_, CO_2_ and C_2_H_4_ have the greatest impact on the quality of sweet cherry. Literature shows that O_2_ and CO_2_ concentrations may stimulate or reduce the respiration rate of fruit cells, CO_2_ in the microenvironment does not directly inhibit the production of ethylene but may bind with some metal containing enzymes (such as catalase) in the sweet cherries fruits and affect the physiological indicators of the fruit. O_2_ is necessary for ethylene production, which may be inhibited at low O_2_ concentrations [37]. Consequently, these factors are closely related to sweet cherry quality [38]. As a plant hormone, C_2_H_4_ plays an important role in fruit ripening and aging, and it has high response characteristics even for non-respiratory fruits such as sweet cherry. However, according to previous studies, there is only a low level of the Period I in the process of ripening and senescence of non-respiratory climacteric fruits such as sweet cherry [39]. Therefore, the C_2_H_4_ content in the sweet cherry transportation microenvironment is very low, which requires high precision and resolution for the C_2_H_4_ sensor.

The gas sensors were tested and calibrated respectively after assembled. Through pre-experiment, the sensors were tested to check the input and output characteristics, to determine whether the errors were within the allowable range and provide a reference for subsequent data analysis. The sensors are tested and calibrated with standard gas at laboratory temperature of 25 °C and relative humidity of 30%. Test point parameters are shown in Table 1. Among three groups of different O_2_ concentration, the maximum absolute error was 0.08%vol; the CO_2_ sensor was tested under five groups of different CO_2_ concentration, and the maximum absolute error was 0.0132%; the C_2_H_4_ sensor was tested under four groups of different C_2_H_4_ concentration, and the maximum absolute error was 1.01%. It can be seen from the static characteristics (Figure 3) that: the static calibration curves of the three gas sensors are approximately straight lines, indicating that these have a good linear relationship. That is, the linear error of the sensor is small, which means that the actual input-output curve error of the sensor is small; besides, the sensitivity of the C_2_H_4_ sensor is the highest reaching 161.37 V/%vol when compared with an O_2_ sensor (0.067 V/%vol) and CO_2_ sensor (0.16 V/%vol), which means the ethylene sensor could monitor the small fluctuations of ethylene content in the microenvironment.

#### 2.1.2. Software Design

The software of the Monitoring terminal was edited by Keil Uvison4 software, and the development language was C language. The main working flow of the software as shown in Figure 4.

Step 1: Initialization. The first step of the main program is an initialization, including initializing the clock, Electrically Erasable Programmable read only memory (EEPROM), main function, etc., in order to ensure that each unit works properly.

Step 2: Data acquisition. By performing an interrupt operation, the main program sends a signal acquisition command to each sensor node, and the interrupt program begins execution. Through the main control chip, each device node starts collecting micro-environment signals by received a high potential. If the signal acquisition is completed, it returns directly to the main program; if the signal acquisition fails, the loop statement is executed, and the acquisition is performed again until the complete signal is acquired.

Step 3: Data reception. After each sensor node collects the signal, it returns to the data buffer (RAM) through the data bus serial port, and the counter increases by one, program execution judgment function, actively judge whether there is data in the buffer, if there is, ready to read and process one by one;

Step 4: Reading the data. The program automatically reads the unprocessed data from the receive buffer and stores it in read-only memory (ROM). The read counter increments by one. After the data is stored in ROM, users or staff can use a personal computer or laptop to read the data in the storage chip. Data cable is used to connect the universal serial bus (USB) module of computer and single chip microcomputer, and *STC* chip control software is used to read data directly.

Step 5: After the data was sent to RAM and stored in ROM, the sensor node begins to sleep and waits for the next collection instruction cycle sent by the main control chip.

#### 2.1.3. Construction of Freshness Prediction Model

The goal of the freshness prediction model is to establish the coupling relationship between gas content and the quality of sweet cherries. Consequently, the quality of sweet cherries can be quantified, which is more specific and intuitive [40]. Since the freshness of sweet cherries is related to many parameters in the microenvironment, and these parameters also affect each other, the relationship between freshness and key parameters is non-linear. Therefore, it is necessary to select a suitable method to establish a sweet cherry freshness prediction model in the express logistics, thereby achieving high accuracy [28].

A BP neural network is neural network using an error backpropagation training algorithm. It has a multi-layer feedforward network with a hidden layer, which provided a higher prediction accuracy, adaptability, and robustness compared with the traditional linear model [41]. The basic principle of the BP neural network is the grades decline method, and the core idea is to adjust the weight value and threshold value to minimize the total error [42]. The BP neural network can implement most of the complex nonlinear mapping, which makes it particularly suitable for solving complex problems. The work process of the BP neural network is mainly divided into two steps: (1) the forward propagation of the signal. The input signal is processed layer by layer from the input layer through the hidden layer and finally reaches the output layer. The state of each layer of neural units only affects the state of the next layer of neural units; (2) the backpropagation of errors. If the expected value is not obtained at the output layer, it is transferred to the backpropagation, and the error signal is returned along the original connection channel. By modifying the weight and threshold of each layer of neurons, the error signal is minimized. Through these two steps, the error of the BP neural network is gradually approached to the expected value. The conceptual structure of the prediction model based on BP neural network is shown in Figure 5.

The freshness prediction model is formed by connecting a plurality of neurons by certain rules, and mainly includes an input layer, implicit layers (intermediate layers), and an output layer. The number of neurons in the input layer is the same as the dimension of the input data. The three gas contents (O_2_, CO_2_ and C_2_H_4_) in the sweet cherry microenvironment are considered as the input, and the output layer is the freshness of the sweet cherry. $w_{ki}$ and $w_{ij}$ are the weighting coefficients of the output layer and the hidden layer, respectively. The establishment steps of the prediction model based on the BP neural network are as follows:

The first step is to determine the structure of the prediction model: a neural network is generally composed of an input layer, hidden layer and output layer, and the theory and method on how to determine the number of hidden layers has not been defined yet. Therefore, according to the following empirical formula [40], the hidden layer ranges from 3 to 12.
(1)$$M=\sqrt{P+T}+a$$
where M represents the number of neurons in the hidden layer; P represents the number of neurons in the input layer; T represents the number of neurons in the output layer; a is a constant, usually between 1 and 10. The BP neural network with different hidden layers was tested, and the results were shown in Table 2. It could be seen that, when the number of hidden layer neurons was different, the iteration and mean square error (MSE) were different. When the number of hidden layers of neurons was 10, although the number of iterations is not the least, the MSE was the smallest, namely the highest precision. Therefore, the number of hidden layers was 10 for the BP neural network.

The second step is to determine the number of nodes in each layer: according to the above analysis, the input layer were the three gas contents, so the number of input nodes is 3; the output layer is freshness, so the number of output nodes is 1.

The third step is data normalization: data normalization not only facilitates the comparison of different parameters but also speeds up the convergence of the training network. Here, the min-max normalization method was used, which could simplify data processing and reduce the load of processing chips compared with the Z-score method. The normalized formula was as follows:(2)$$\varepsilon =\frac{h-h_{min}}{h_{max}-h_{min}}$$
where: ε—Normalized data; h—original data of each gas content; $h_{max}$—maximum value in gas data; $h_{min}$—minimum value in gas data.

The fourth step is to choose the appropriate function for training. In this paper, according to the range of normalized experimental data (0–1), “tansig” and “logsig” functions were selected as transfer functions, and “trainrp” functions were selected as training functions for the freshness prediction model. The maximum iteration number was set as 1000, the learning rate was set as 0.001, the accuracy was set as 0.001, and other parameters were set as default values. In this study, a total of 9000 gas experimental data of sweet cherry during transportation were divided into two parts according to the ratio of 7:2. The first part was used to train the model, and the second part was used to testing.

The MATLAB R2016b math software (version 9.1, MathWorks Inc., Natick, MA, USA) was used for modeling. The construction and training results of the freshness prediction model are shown in Figure 6, which can be seen that: after 87 iterations of the neural network, the error requirement has been met; most of the predicted data fall on the line of the true value, and other points are evenly distributed on both sides of the line. The R-value is also closed to 1, which indicates that the neural network has high accuracy.

### 2.2. Experiment Scenario

The experiment was conducted in Yantai city, Shandong province. The Yantai area is the main production region and distribution center for sweet cherries in China [43]. The “Meizao” variety is a hybrid of Stella and EarlyBudm, which is characterized by easy planting, high yield, storage and transportation resistance. Hence, the “Meizao” variety has been planted in large areas in the Yantai region, and is a commonly traded cherry variety. The experiment was conducted in June which is the peak harvesting time of “Meizao” cherries.

Immediately following their harvest by manual or mechanical, the fruits were sorted and graded to remove fruit damaged by disease and insects, and then immediately transported to the laboratory. The experiment was mainly divided into three groups: ambient temperature group (Group I), ice-added group (Group II) and pre-cooling group (Group III), as shown in Figure 7.

A total of 66 kg of sweet cherries of the same size and maturity (90% maturity) were selected and divided into three groups: I, II, and III. For each group, the sweet cherries was divided into 11 independent portions, in which ten portions were used for the quality test while the eleventh portion was used for monitoring the critical micro-environmental parameters. The weight of each portion was controlled within 2 ± 0.02 kg. Group I was used to simulate ambient temperature transportation; group II was used to simulate the ice-added transportation (two 200 mL ice-bags were placed in each independent packages); Group III was used to simulate the pre-cooling transportation, which was placed in a refrigerated warehouse (temperature was 0–1 °C for the first 12 h). The acquisition frequency of sensors was 27 s/time, and the quality indicators were tested twice a day at 8 am and 5 pm, respectively.

### 2.3. Analysis Method Based on HACCP

The hazard analysis and critical control point (HACCP) is a quality control system that ensures the quality and safety of food by identifying, evaluating, and controlling various hazards in food production, processing, sales, and consumption. It has been widely used in the food industry, especially in the cold chain logistics of fresh fruits or vegetables, meat and seafood [44,45]. The HACCP method was used in this paper to analyze the typical business flow of fresh sweet cherries, including identifying the potential hazards (HA) and critical control points (CCP), determining the appropriate control measures to each critical control point.

### 2.4. Quality Indicators Measurement of Sweet Cherries

In order to evaluate the quality of fresh sweet cherries, critical quality indicators were selected, including chromatic aberration (L*, a*, b*), hardness, pH, and soluble solid content (SSC). Ten random sweet cherries were measured twice a day at 8 am and 5 pm respectively. Detailed parameters of the measurement devices were shown in Table 3.

The chromatic aberration was measured by the CR-410 chromatic aberration analyzer (produced by Konica Minolta Co., Ltd., Tokyo, Japan). RH: Relative humidity. To reduce the error caused by uneven color on the surface of sweet cherries, the peaks on both sides and the middle position of the sweet cherries were measured, and then the three values were averaged to obtain the final value of the chromatic aberration (L*, a*, b*). Among them, the values of the brightness L*, the red value a*, and the yellow value b* were directly obtained by measurement, but to integrate the effects of the three parameters, the value of the total chromatic aberration ΔE was calculated by the following formula [46]:(3)$$\Delta E=\sqrt{\left[{\left(L*\right)}^{2}+{\left(a*\right)}^{2}+{\left(b*\right)}^{2}\right]}$$

The hardness was measured by FHT-05 Hardness Tester (produced by Landtek Co., Ltd., Guangzhou, China). The peaks on both sides of the sweet cherry were measured, and the penetration speed was controlled at 0.5mm/s. The maximum value was not recorded until the peel of sweet cherries was punctured, then the average value of the two measurements was obtained.

The pH was measured by Testo 205 (produced by Testo SE & Co. KgaA). Before measurement, the pH probe was washed with distilled water and cleaned with a dust-free paper, then the probe was stabbed into the two peaks of fruits, and the data was recorded after the displayed value was stable. The final value was the average value of the two measurements.

The soluble solid content (SSC) was measured by a hand-held digital refractometer (manufactured by ATAGO Corporation). Before measurement, the prism part on the refractometer was washed with distilled water, then the dust-free paper was used to clean the remaining water, then 2–3 drops of juice were apply to the surface of the prism. The displayed value was recorded after stable. Each fruit was tested twice, and the final value is taken as the average of two measurements.

## 3. Results and Discussions

### 3.1. Business Flow Analysis for Fresh Sweet Cherries Based on HACCP

Through the relevant literature and field research in Yantai City, a typical business flow (from harvesting to sale) of sweet cherries in China was summarized and the hazard points, as well as corresponding control measures were analyzed based on Hazard Analysis and Critical Control Point (HACCP), the business flow and hazard analysis and critical control points could be seen in Figure 8 and Table 4, respectively.

Step Ⅰ: Harvesting (production link). Sweet cherries generally ripen from May to July, and picking is usually carried on a sunny morning when the outdoor temperature is around 25 °C. The purpose is to keep the temperature of sweet cherries at a low level, thereby reducing the risks of quality deteriorates due to high temperature.

Step Ⅱ: Sorting and grading (production link). After harvesting, the sweet cherries are sorted and graded by workers or machines within 2 h. The purpose is to remove fruits that are broken or damaged by pests and diseases, and put together fruits of the same size and maturity.

Step Ⅲ: Packaging (production link). A certain weight of sweet cherries is put in a polystyrene foam plastic box and sealed with tape, and then waiting for the next step of processing or directly shipped to the local market for sale.

Step Ⅳ (Model 1): Local sales (circulation link). In order to meet the needs of the local market, some sweet cherries are sold locally after packaging, which is usually completed within 1–2 h;

Step Ⅳ (Model 2): Short-distance transportation (circulation link). To meet the needs of the surrounding market, sweet cherries are transported to the nearby towns through light-van for sale, which is generally completed within 3–5 h;

Step Ⅳ (Model 3): Express transportation (circulation link). With the development of the Internet economy and express industry in China, online sales account for an increasing share in sweet cherry sales models. Sweet cherry is firstly packaged and then distributed to different cities through express logistics, which usually requires an average of 6–72 h. To ensure the taste of sweet cherries and reduce the quality loss during the transportation process, ice bags are usually added in the express package;

Step Ⅳ (Model 4): Long-distance transportation (circulation link). Some sweet cherries are immediately sent to the cold storage for pre-cooling after packaging. The pre-cooling temperature is generally 0–1 °C and the pre-cooling time is around 12 h, which can fully dissipate the field heat, reduce the core temperature of fruits, slow down the respiration rate and extend the shelf life of sweet cherries. After the full pre-cooling, the sweet cherry is transported to distant cities by cold chain trucks immediately, and the transportation time generally no less than 72 h;

Step Ⅴ: Terminal clients (consumption link). Terminal clients include wholesalers, retailers, and consumers. Lots of refrigerated shelves are used in wholesalers and retailers (mainly supermarkets) to ensure that the temperature and relative humidity are within an appropriate range for reducing quality loss. The express transportation mode directly delivers sweet cherries to consumers, completing the entire transportation chain structure.

### 3.2. Mechanism of Quality Change of Fresh Sweet Cherries

The sweet cherries are still a living organism after harvesting, which means respiration, transpiration and other metabolic activities are still going on [47,48]. Under the influence of various critical micro-environmental parameters (temperature, relative humidity, O_2_, CO_2_ and C_2_H_4_), the quality of sweet cherries are changing constantly during transportation. The mechanism of quality changes in express logistics was analyzed in Figure 9.

#### 3.2.1. Effect of Temperature and Relative Humidity on the Quality of Sweet Cherries

The temperature and relative humidity have an important effect on the quality of sweet cherries in express logistics. During this process, temperature directly affects a series of physiological activities of sweet cherries, such as respiration (aerobic respiration and anaerobic respiration), transpiration, and biosynthesis activity (such as ethylene synthesis). The relative humidity directly affects the water loss and inhibits hydrolysis as well as reduces the respiration intensity of sweet cherries [49]. The temperature and relative humidity further affect decay, quality parameters (hardness, water content, etc.), and microbial growth and reproduction, which eventually affects the fruits’ appearance, commodity value and nutritional value. B. W. et al. (2014) found that Regina sweet cherries could still maintain good sensory quality after stored at 0 °C for 35 days, which meant that the suitable environmental temperature and relative humidity could keep sweet cherries in good quality [50].

#### 3.2.2. Effect of Gas Content on Sweet Cherry Quality

The gas content in the express logistics microenvironment also has an important effect on the quality of sweet cherries. R. Cozzolino et al. (2019) found that in low oxygen microenvironment (about 1%), sweet cherries were sensitive to CO_2_ accumulation (20%), which showed by the increase in respiration rate (aerobic and anaerobic respiration), biosynthesis of fermentative volatile metabolites, and sensory perception of off-odors [51]. Although it is believed that sweet cherries are non-respiratory fruits, according to the research of Tianmei Jiang et al. (2011), the ethylene also had an important influence on the quality of non-respiratory saltation fruits, which could promote the color change from green to yellow and relieve hardness reduction [39].

When the microenvironment is in a state of high oxygen and low carbon dioxide, the respiratory rate of sweet cherries is high. The oxygen in the microenvironment was consumed, and the carbohydrates and organic acids were converted into energy, carbon dioxide, and water by enzymatic reaction [52], which causes the quality loss and consumption of nutrient, at the same time micro-environmental temperature, relative humidity and carbon dioxide from rising rapidly. With the decrease of oxygen and the increase of carbon dioxide, the respiration rate of sweet cherries is gradually inhibited, which reduces the consumption of nutrients. In this way, which could help to reduce the quality loss and prolong the shelf life. Overall, the changes of these gases content directly or indirectly affect the fruits’ appearance, nutritional value and commercial value of sweet cherries, which could further affect the market price and reduce consumers’ purchase intention. Therefore, in order to reduce quality loss and strengthen management level in express logistics, it is necessary to closely monitor the changes of these gases content in the transport microenvironment.

### 3.3. Monitoring Data Analysis

#### 3.3.1. Temperature and Relative Humidity Data Analysis

The changes of temperature and relative humidity in the microenvironment of sweet cherries in express logistics were shown in Figure 10.

As can be seen from Figure 10, the trends of the three experimental groups are the same. More specifically, both temperature and relative humidity were rising over time. As sweet cherries are still undergoing respiration and transpiration, which releases temperature and moisture, causing the temperature and relative humidity in the microenvironment to rise rapidly. The temperature of the ambient temperature group was raised in the first 1500 min, and then the temperature was stable at 27.6 °C; the temperature of the ice-added group and the pre-cooling group showed a sharp decline in the micro-environment of the sweet cherries due to the external cold source. The temperature of the ice-added group dropped to a minimum of 16.7 °C at about 75 min; because of cold storage (the cooling effect was better than ice-added group), within two hours, the microenvironment temperature dropped to 9.9 °C at about 124 min. The temperature of both the ice-added group and the pre-cooling group gradually rose to the same level as the ambient temperature group (about 27 °C). From Table 5, the mean temperature of these three groups was 24.8 °C, 22.2 °C, and 18.5 °C respectively.

The relative humidity of the three groups reached 90% in a short period and then remained stable in the range of 90% to 98%. The ice-added group spent the longest time to reach 90% relative humidity, for about 6.8 h. The pre-cooling group showed the fastest speed, the relative humidity of the microenvironment had reached 90% after 2.2 h. The ascent speed of ambient temperature group was middle, the relative humidity reached 90% after 3.9 h. According to previous studies, the most suitable temperature and relative humidity for the storage of sweet cherries were 0–1 °C and 90–95% relative humidity. From the Table 6, it is clear that the mean relative humidity of the ice-added group and pre-cooling group (76.15% and 74.60% respectively) were lower than that of the ambient temperature group (83.6).

#### 3.3.2. Gas content Data Analysis

The variation of gases content in the microenvironment of sweet cherries under three different express transportation modes is shown in Figure 11. Although the environmental parameters of the three transportation modes were different, the general trend was similar, in which the O_2_ content was gradually decreasing, the CO_2_ and C_2_H_4_ content were gradually increased. This phenomenon is closely related to the metabolism activities of sweet cherries (including aerobic and anaerobic respiration). The respiration activity of sweet cherries consumes the O_2_ and emits CO_2_ and C_2_H_4_ during the whole express transportation.

In terms of O_2_ content, the O_2_ consumption rate of the ambient temperature treatment was the fastest (about 0.038%/min). In the S1 and S2 phase, the O_2_ content dropped rapidly. More specifically, the ice-added group showed a higher rate of O_2_ consumption than that of pre-cooling treatment in the S1 phase, while the rate was exactly the opposite in the S2 phase. The O_2_ content remained at a stable level (about 2.61%, 9.94%, and 8.72% respectively at the end).

As for CO_2_ content, the CO_2_ content in the ambient temperature group indicated the fastest increase compared with the other two groups. In the S1 phase, the growth rate of CO_2_ in the ice-added group (0.0104%/min) is greater than that in the pre-cooling group (0.0098%/min), while an opposite trend was presented in the S2 phase, which also corresponded to the consumption of O_2_ content. Finally, in the S3 phase, the CO_2_ content leveled off at 10.559%, 8.362% and 9.338% respectively.

In general, ethylene content maintained an upward trend during the experiment period, but the content and variation range was very small. The three experimental groups reached the maximum (about 0.0029%, 0.00152% and 0.00201% respectively) in the late stage of the S2 phase, and then the ethylene content decreased slightly in the S3 phase. Since the sweet cherries are considered as non-climacteric fruits, and the respiration rate has been declining after harvesting, there is no obvious increase in C_2_H_4_ content after harvesting compared with climacteric fruits (such as apple and banana). The possible reason was that the O_2_ content in the sealed box was too low to continue the aerobic respiration of sweet cherries, and anaerobic respiration became the main mode of respiration to provide energy. When the O_2_ content in the microenvironment less than 1%, the rate of anaerobic respiration was accelerated, the content of ethanol and acetaldehyde in the fruit increased rapidly, and the flavor of the sweet cherry also decreased.

From the perspective of temperature and relative humidity changes, the pre-cooling treatment could more effectively suppress the temperature increase compared to the ambient treatment or ice-added treatment, although the high relative humidity could be caused at the same time. From the perspective of the changes of the three gases, when the transportation time is less than 10 h (about 600 min), the pre-cooling treatment has the most obvious inhibitory effect on the physiological metabolism (such as respiration) of sweet cherries; while the transportation time is greater than 10 h, adding ice packs in the express package may have a better effect. Since the changes in gas content were also affected by the temperature and relative humidity, the analysis from the perspective of the change of gas content was more guiding significant.

#### 3.3.3. Quality Data Analysis

The quality indicators change of sweet cherries during the express logistics could be seen in Figure 12. As time went by, the color of sweet cherries changed from light to dark, specifically from bright red to deep red (the value of L* ranged from 29.41 to 28.70); the value of pH changed from 3.58 to 4.40, which meant the taste of sweet cherries became unpalatable and the sweet cherries became soften and lacked cohesion (the value of hardness changed from 5.62 to 4.30); the value of SSC changed from 17.06% to 13.27%. Meanwhile, the stem of the sweet cherries became withered and the color changed from green to brown, which will greatly affect consumers’ purchase intention. Through the comparison table of freshness and quality parameters, stakeholders could assess the quality of sweet cherries according to the predicted value from the prediction model, which could also provide a basis for relevant decision-making.

### 3.4. Evaluation of Freshness Prediction Model

According to the construction steps of the prediction model in section two, the O_2_, CO_2_, and C_2_H_4_ content were used as the input of the prediction model. In total, 7000 sets of data were selected for training the prediction model, and the remaining 2000 sets of data were used for testing and evaluation of the predictive model. The predicted and actual values of freshness in three different express logistics conditions were obtained, and the absolute and relative errors between the predicted and true values were shown in Table 5.

As can be seen from Table 5, the prediction model showed good accuracy in predicting the freshness of sweet cherries in the ice-added group and pre-cooling group, while the prediction accuracy is slightly worse for the ambient temperature group. The absolute errors ranged from −0.254 to 0.273, and the relative errors were controlled within 10% and could meet the basic prediction accuracy.

### 3.5. Evaluation of the DMQAS

The DMQAS were evaluated and discussed through questionnaires and interviews with stakeholders in the whole process, and relevant feedback was collected for promoting the improvement of DMQAS. After the questionnaire survey and interviews, it was found that the DMQAS could real-time accurately monitor critical parameters in the express logistics microenvironment of fresh sweet cherries. Table 7 provided a comprehensive comparison of the traditional monitoring system and the DMQAS in detail. The evaluation Table 7 illustrated that: the proposed DMQAS can accurately monitor more parameters in real-time (O_2_, CO_2_, and C_2_H_4_) compared with the traditional monitoring system (only monitoring of temperature and relative humidity), the accuracy of each sensing unit was higher, response time and recovery time were shorter, and the DMQAS could accurately predict the freshness of sweet cherries through prediction model based on BP neural network, which could provide a reliable reference to stakeholders for decision-making and improving express logistics management. Overall, the quality loss rate decreased by 10%–15%, and the market price of fresh sweet cherries increased by about 50% (the market price increased from about 40 yuan/kg to 60 yuan/kg), which greatly reduced the quality and economy loss.

## 4. Conclusions

This paper presents dynamic monitoring and quality assessment system (DMQAS) based on multi-sensors, for reducing the quality loss of fresh sweet cherries in the express logistics. The system can monitor key micro-environmental parameters (temperature, relative humidity, O_2_, CO_2_, and C_2_H_4_) in real-time during express logistics, and can also accurately predict the freshness of sweet cherries (the relative error can be controlled within 10%). Through the data analysis, it is found that the pre-cooling treatment is suitable for a short duration (<10 h) express logistics, and the ice-added treatment is more effective for a long duration (>10 h) express logistics. Combined with the quality parameters (hardness, SSC, pH, and chromatic aberration), the DMQAS could provide theoretical reference for timely decision-making and logistics management, the quality loss of the sweet cherry was reduced to 15% from 25%–30%, and the market price of sweet cherry increased from 40 yuan/kg up to 60 yuan/kg via evaluation.

Although the DMQAS can meet the basic needs of the actual process, the sensors currently used are rigid and bulky, which may cause secondary damage to fruits. In the future, the DMQAS will integrate flexible micro-sensors, which could be adhered to the express packaging to achieve non-destructive monitoring. Future research will also focus on adding new sensing units to apply the DMQAS to other product logistics scenarios or adding dynamic real-time adjustment devices to achieve the function of remote real-time quality control.

## Figures and Tables

**Figure 1 foods-09-00602-f001:**
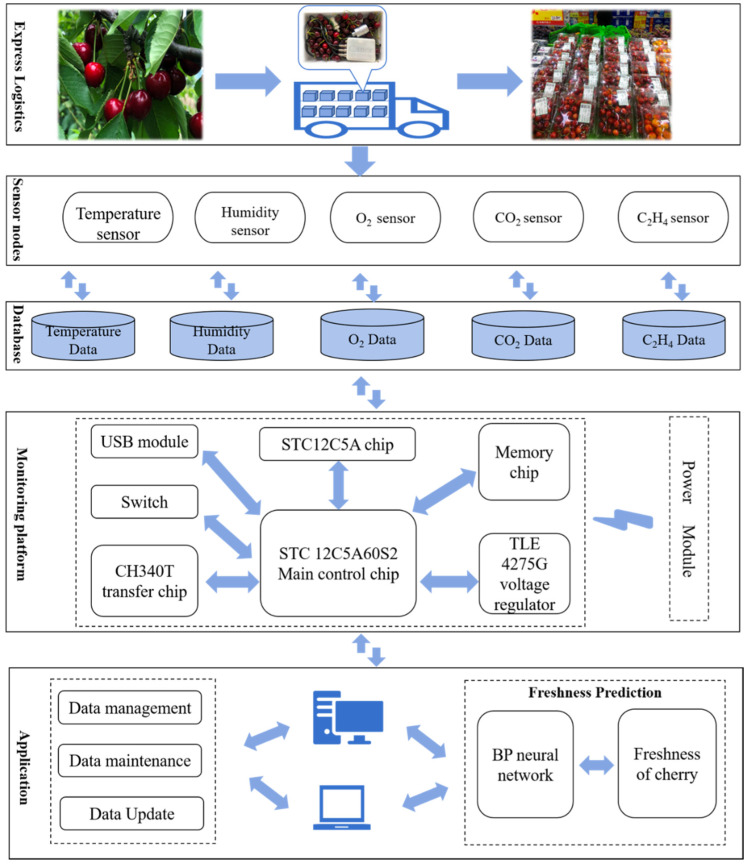
The framework and logical relationships of the dynamic monitoring and quality assessment system (DMQAS).

**Figure 2 foods-09-00602-f002:**
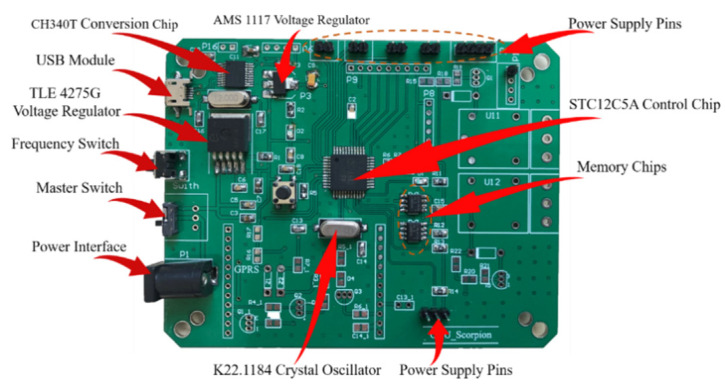
The physical architecture of the control platform based on single chip microcomputer.

**Figure 3 foods-09-00602-f003:**
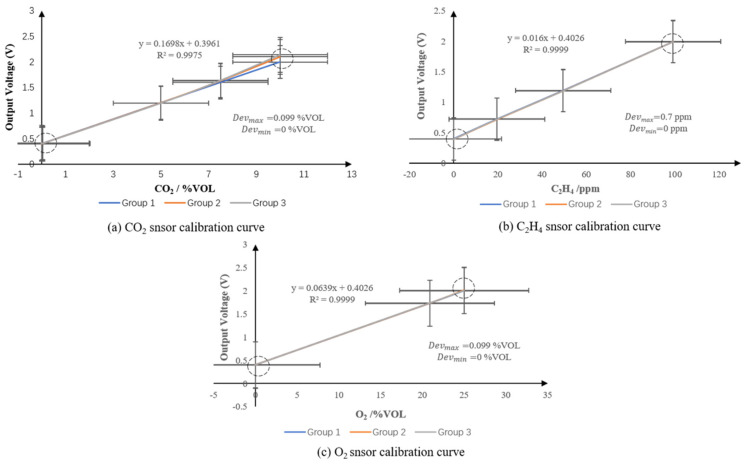
The static calibration curve and analyses of gas sensors. (**a**) CO_2_ snsor calibration curve; (**b**) C_2_H_4_ snsor calibration curve; (**c**) O_2_ snsor calibration curve.

**Figure 4 foods-09-00602-f004:**
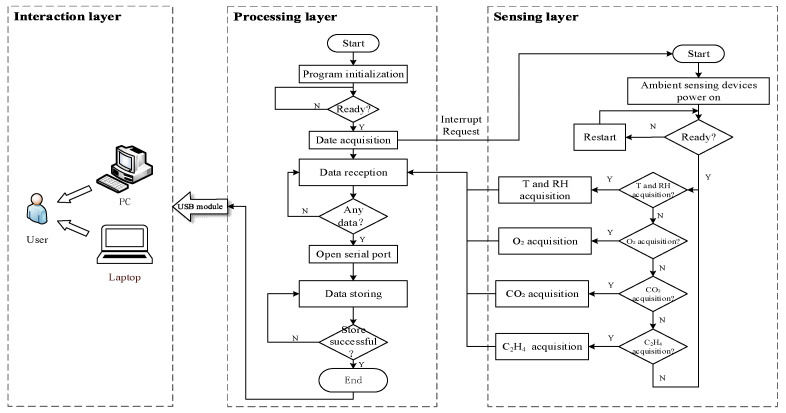
The processing flow of multi-sensors signal acquisition.

**Figure 5 foods-09-00602-f005:**
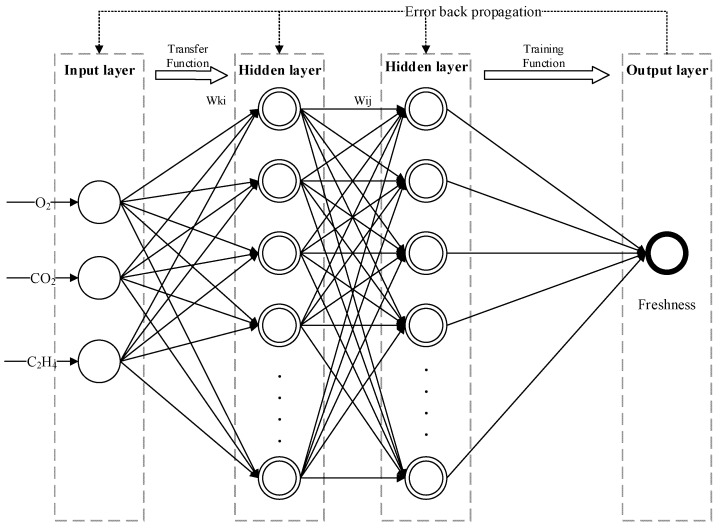
The conceptual structure of freshness prediction model based on back propagation (BP) neural network.

**Figure 6 foods-09-00602-f006:**
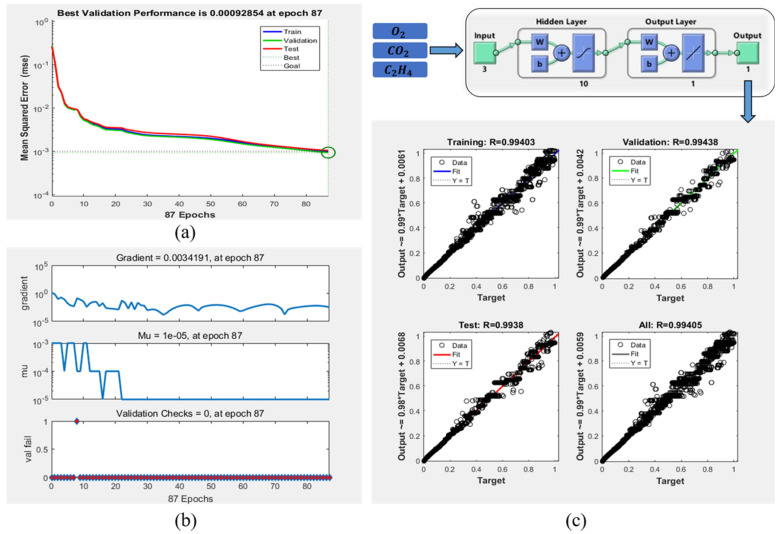
Structure and performance of the prediction model based on BP neural network. (**a**) The best validation performance at 87epochs; (**b**) the gradient and validation fail at 87 epochs; (**c**) the fit of the output value and target.

**Figure 7 foods-09-00602-f007:**
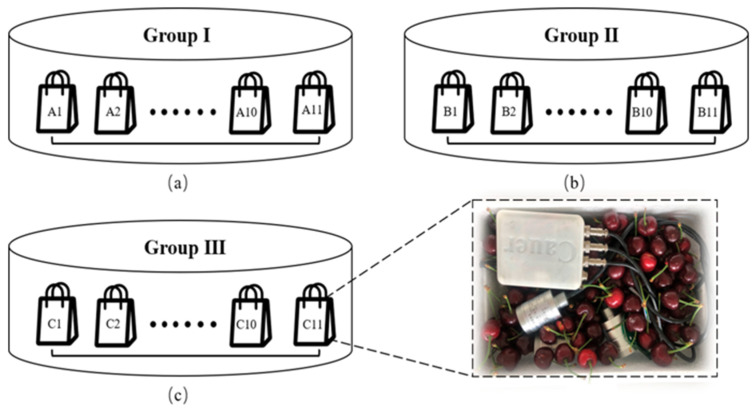
Simulation experiment of sweet cherries in express logistics transportation. (**a**) ambient temperature group; (**b**) ice-added group; (**c**) pre-cooling group.

**Figure 8 foods-09-00602-f008:**
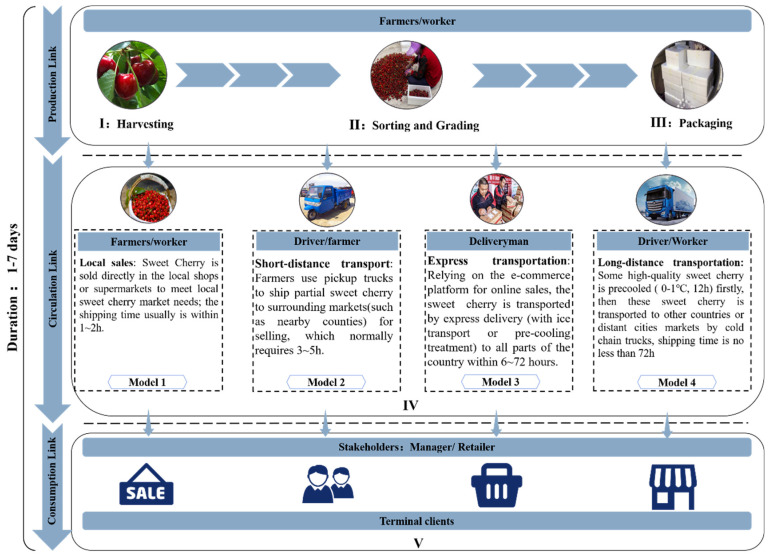
The typical business flow of the sweet cherries industry.

**Figure 9 foods-09-00602-f009:**
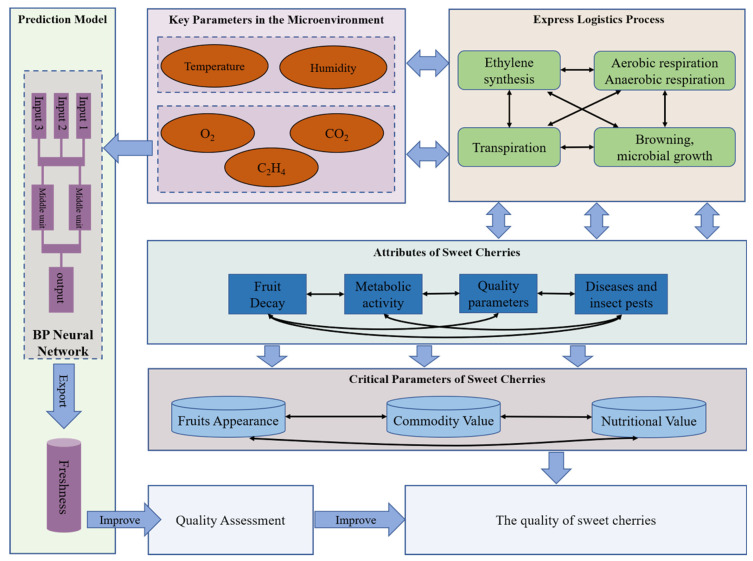
Mechanism of Quality Change of Sweet Cherries in the Express Logistics.

**Figure 10 foods-09-00602-f010:**
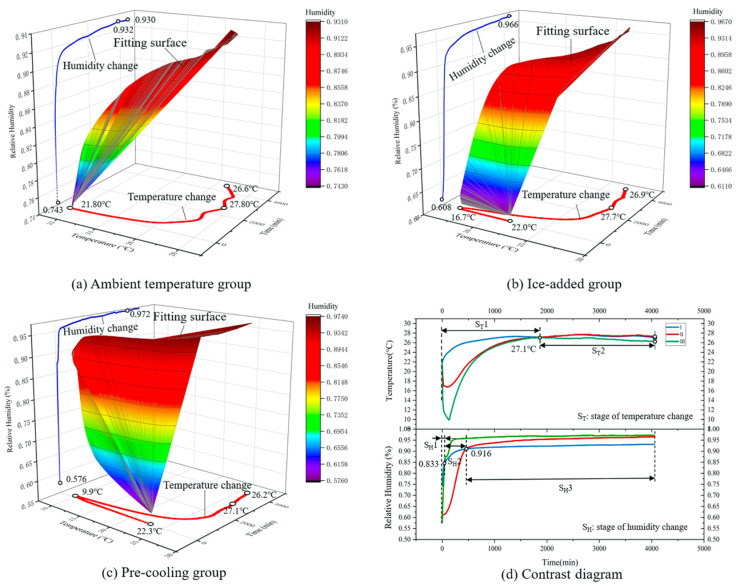
Changes of temperature and relative humidity in express logistics. (**a**) Ambient temperature group; (**b**) Ice-added group; (**c**) Pre-cooling group; (**d**) Contrast diagram.

**Figure 11 foods-09-00602-f011:**
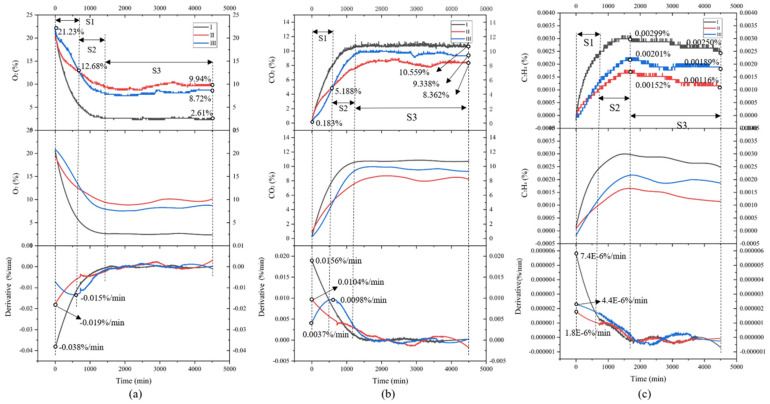
Changes of the microenvironment gas content in the express transportation. Note: I represents the ambient temperature group; II represents the ice-added group; III represents the pre-cooling group. (**a**–**c**) three different express transportation modes.

**Figure 12 foods-09-00602-f012:**
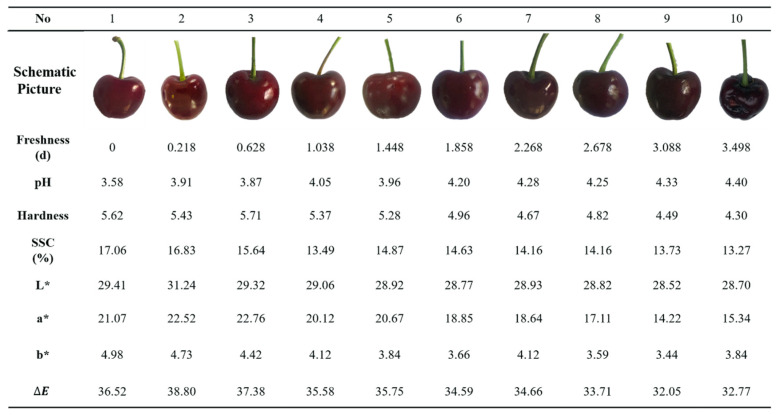
Changes of quality indicators of sweet cherries in express logistics.

**Table 1 foods-09-00602-t001:** The calibration data and error analysis of gas sensors.

Type		1	2	3	4	5	Max Error
O_2_ sensor (%vol)	Sv	0	20.9	25	-	-	0.08%
	Mv	0.00	20.89	25.02	-	-	
	Ov	0.4	1.732	2.005	-	-	
CO_2_ sensor (%vol)	Sv	0	0.040	5.000	7.500	10.000	0.0132%
	Mv	0.00	0.041	4.996	7.599	9.998	
	Ov	0.4	0.407	1.201	1.600	2.000	
C_2_H_4_ sensor (%vol)	Sv	0	19.6	49.5	99.2	-	1.01%
	Mv	0.00	19.8	49.9	99.9	-	
	Ov	0.399	0.723	1.194	1.996	-	

Notes: Sv: standard values (%vol); Mv: measured values (%vol); Ov: output values (V).

**Table 2 foods-09-00602-t002:** Comparison of training results of different numbers of hidden layers.

Hidden Layers	Epoch	MSE	Hidden Layers	Epoch	MSE
3	101	1.30 × 10^−3^	8	145	9.96 × 10^−4^
4	25	3.48 × 10^−3^	9	80	1.00 × 10^−3^
5	115	9.94 × 10^−4^	10	87	9.92 × 10^−4^
6	247	1.02 × 10^−3^	11	168	9.98 × 10^−4^
7	165	9.96 × 10^−4^	12	116	9.99 × 10^−4^

**Table 3 foods-09-00602-t003:** Specifications of quality indicators measurement devices.

No.	Quality Indicator	Device	Operating Environment	Accuracy	Measurement/Display Range	Others
1	Chromatic aberration	KONICA MINOLTA CR-400	T: 0–40 °C RH: <85%	-	Y: 0.01–160.00%	Standard deviation ΔE∗ab<0.07
2	Hardness	FHT-05	T: 0–45 °C RH: <90%	±0.01 kg f	0.2–5.0 kg f	Probe insertion depth: 10 mm
3	pH	Testo 205	T: −20–70 °C RH: <85%	±0.02 pH	0–14 pH	-
4	Soluble Solid Content	ATAGO PAL-1	T: 10–40 °C RH: None	±0.2% Brix	0.0–53.0%	-

**Table 4 foods-09-00602-t004:** Business flow analysis of sweet cherries based on hazard analysis and critical control point (HACCP).

Stage	Critical Control Point (CCP)	Hazard Analysis (HA)	Stakeholders	Control Measures
1	Harvesting	Pests and diseases, immature, decay, pesticide residue.	Farmers, picking staff	The picking process should be standardized to avoid harvesting substandard sweet cherries.
2	Sorting, grading, packaging	Non-standard sorting and grading methods, poor operating environment, unreasonable packaging	Farmers, workers	The standardized sorting and grading methods should be adopted, clean and hygienic working environment should be guaranteed.
3	Pre-cooling	No complete pre-cooling, cross-infection.	Cold storage management staff	The complete pre-cooling and clean pre-cooling environment should be ensured.
4	Short-distance transportation	Transportation factors such as packaging, vibration, temperature, etc.	Transport personnel	The suitable packaging method and temperature control measures could be provided, violent vibration should be avoided.
5	Express transportation	Unsuitable storage and transportation, unstandardized operations, lack of process management	Express companies	Reasonable storage and transportation condition should be provided, standardized operations and process management should also be emphasized.
6	Long-distance transportation	The freight density, transportation condition, haulage time.	Transport personnel	Reasonable loading density, critical microenvironment, transportation time should be taken into consideration.
7	Sales or Display	Improper temperature or relative humidity control cause browning or rotting.	Supermarket Salesman Saleswomen	The sales or display environment should be controlled in a suitable range.

**Table 5 foods-09-00602-t005:** Comparison analysis of the predicted and actual freshness.

Freshness (day)	Group Ⅰ			Group Ⅱ			Group Ⅲ		
	Predictive Value	Absolute Error	Relative Error	Predictive Value	Absolute Error	Relative Error	Predictive Value	Absolute Error	Relative Error
2.499	2.334	−0.165	−0.066	2.461	−0.038	−0.015	2.502	0.003	0.001
2.549	2.471	−0.078	−0.031	2.393	−0.156	−0.061	2.585	0.036	0.014
2.599	2.589	−0.010	−0.004	2.445	−0.154	−0.059	2.616	0.017	0.007
2.649	2.726	0.077	0.029	2.463	−0.186	−0.070	2.589	−0.060	−0.023
2.699	2.589	−0.110	−0.041	2.445	−0.254	−0.094	2.616	−0.083	−0.031
2.749	3.002	0.253	0.092	2.579	−0.170	−0.062	2.947	0.198	0.072
2.799	3.072	0.273	0.098	2.585	−0.214	−0.076	2.933	0.134	0.048

**Table 6 foods-09-00602-t006:** The details of the change of temperature and relative humidity.

Groups	Temperature				Relative Humidity			
	Maximum Value (°C)	Minimum Value (°C)	Mean (°C)	Maximum Rate (°C /min)	Maximum Value (%)	Minimum Value (%)	Mean (%)	Maximum Rate (%/min)
Ⅰ	27.80	21.80	24.80	0.05	93.20	74.00	83.60	6.02
Ⅱ	27.70	16.70	22.20	0.56	96.60	60.80	78.70	4.78
Ⅲ	27.10	9.90	18.50	0.89	97.20	57.60	77.40	7.90

**Table 7 foods-09-00602-t007:** Evaluation and analysis of the DMQAS.

System Performance Indicators	Multi-Sensors Micro-Environmental Monitoring	Freshness Prediction	Range/Accuracy of the DMQAS				Response of the DMQAS				Quality Control	
			Temperature and Relative Humidity Monitoring	O_2_ Sensor	CO_2_ Sensor	C_2_H_4_ Sensor	Temperature and Relative Humidity Monitoring	O_2_ Sensor	CO_2_ Sensor	C_2_H_4_ Sensor	Quality Loss	Market Price (RMB)
Traditional work	Temperature, relative humidity	None	Range (T): −40–120 °C Range (RH): 0–99% Accuracy (T): ±1 °C Accuracy (RH): ±0.5%	None	None	None	Response (T): <10 s Recovery (T): <20 s Response (RH): <8 s Recovery (RH): <60 s	None	None	None	25–30%	<40 yuan/kg
Dynamic Monitoring and Quality Assessment System (DMQAS)	Temperature, relative humidity, O_2_, CO_2_, C_2_H_4_.	Relative error <10%	Range (T): −40–80 °C Range (RH): 0–99% Accuracy (T): ±0.5 °C Accuracy (RH): ±0.1%	Range: 0–25%vol Accuracy: ±0.1%vol	Range: 0–10%vol Accuracy: ±0.01%vol	Range: 0–100 ppm Accuracy: ±0.1 ppm	Response (T): <6 s Recovery (T): <20 s Response (RH): <5 s Recovery (RH): <60 s	Response: <20 s Recovery: <60 s	Response: <30 s Recovery: <30 s	Response: <30 s Recovery: <60 s	<15%	>60 yuan/kg
Advantages	More comprehensive critical micro-environmental parameters could be monitored.	The freshness of sweet cherries could be predicted in DMQAS.	Better accuracy of temperature, relative humidity, O_2_, CO_2_, and C_2_H_4_ monitoring.				The faster response and shorter recovery time could be achieved in DMQAS.				Less quality loss and higher market price.	

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
