# Peer review of "Multi-Sensors Enabled Dynamic Monitoring and Quality Assessment System (DMQAS) of Sweet Cherry in Express Logistics"

_foods, 2020, doi:10.3390/foods9050602_

Round 1
Reviewer 1 Report
The paper deals with dynamic monitoring and quality assessment system (DMQAS) based on the multi-sensors technology. The proposed system can monitor the key parameters such as temperature, relative humidity, O2, CO2, and C2H4 content during the express logistics. A freshness prediction model based on BP neural network was built to predict the freshness of sweet cherry according to the microenvironment gas content.
The chosen topic is interesting, however, from my point of view it is too applied in nature and I do not see any research and as well as there is a lack of novelty. Everything that is discussed in the article is already known. There is no any literature review of other solution and techniques, other approaches. Something that distinguishes the chosen approach from others.
Author Response
Dear Reviewers,
Thanks for your comments concerning our manuscript entitled “Multi-Sensors enabled Dynamic Monitoring and Quality Assessment System (DMQAS) of Sweet Cherry in Express Logistics”, (manuscript ID: foods-789646). Those comments are all valuable and very helpful for revising and improving our paper, as well as the important guiding significance to our researches. We have studied comments carefully and have made correction, we hope the correction will meet with approval.
Yours sincerely.
Wang Xiang, On behalf of all of authors

Reviewer 2 Report
April 22, 2020
Journal: Foods
Title: Multi-Sensors enabled Dynamic Monitoring and Quality Assessment System (DMQAS) of Sweet Cherry in Express Logistics
Authors: Xiaoshuan Zhanga, Xuepei Wanga, ShaohuaXingb,YunfeiMaa, Xiang Wanga
Dear Editor,
The authors tested the sweet cherry quality using dynamic monitoring and quality assessment system (DMQAS), with the aid of the software processing and freshness prediction model. Their investigations showed that the quality of sweet cherries is easily degraded in the process of express transportation, which poses a huge risk to the quality and safety of sweet cherries.
Comments to authors:
- The authors could add the list of abbreviation.
- The authors would add the novelty and originality of the study.
- Aim of the study is not clear
- Could you summarize the nutritional value and biological importance of cherry in the introduction
- Does DMQAS monitor the nutritional value and chemical composition of the raw materials? Could you give more details?
- "Besides, there are large differences between cold chain logistics and express logistics. which makes it difficult to apply existing monitoring technologies in cold chain logistics to express logistics."; could you explain in more details?
- "Monitoring platform design, sensing acquisition hardware design "; could you remove these sections
- The tests should be triplicated for each sensor. In this regard, is this supposed to be a method/technique paper?
- Mention the accurate results of variation in temperature, relative humidity, Oâ‚‚, COâ‚‚ and C2H4 as detected.
- In sensing acquisition hardware design section, could you add the effect of O2, CO2 on ethylene production in sweet cherry
- The authors could describe the correlation between expiration rate and the extension of the shelf life of sweet cherry
- The quality of table 4 is very low
- Could you improve the quality of all figures, most of them unclear?
- In regards to express companies, kindly describe the control before and after shipping to grantee the quality during long distance with long time.
- Author can suggest/discuss which method will be more effective in transportation.
- The authors could benefit from the following published paper:
Rasheed, D.M., Porzel, A., Frolov, A., El Seedi, H.R., Wessjohann, L.A. and Farag, M.A., 2018. Comparative analysis of Hibiscus sabdariffa (roselle) hot and cold extracts in respect to their potential for α-glucosidase inhibition. Food Chemistry, 250, pp.236-244.
Author Response

(The authors gave the same response as above.)

Reviewer 3 Report
This is an interesting paper, even though I believe it better fits in the journal 'sensors', I suppose it is okay for this journal too.
The authors need to spend quite a bit of time improving their English language expression. The tenses (past, current and future) are constantly mixed up, this needs to be fixed.
I have no issues with the work, the data or the way it is presented. I do think that the conclusion section is a bit long and should be shortened.
I have included a file of your manuscript. I have written many of my comments in that file. I have stopped making comments about the use of the English language by the time I reached the results section. That does NOT mean that the English is correct from that point onwards.

Author Response

(The authors gave the same response as above.)

Reviewer 4 Report
he article presents an exhaustive procedure for the development of dynamic monitoring and quality assessment system (DMQAS) based on the use of multi-sensors. I have some suggestions for authors to improve their work. These follow the text sequence:
-Abstract
Line 18.''..was controlled within''.
-Introduction and elsewhere
Line 39.'' sweet cherries''.
Line 48. Improve reference format.
Line 64. ''refrigeration''.
Line 77.'' blueberry, tricholoma matsutake,'''.
Line 98 and elsewhere. Delete ''And''. Write: ''In addition, a...''.
Line 102. Figure 1.''logical''.
Line 123. ''In addition, MAX810...''.
Line 169, Figure 3.'' The static....''.
Line 259. h parameter refers to its average value? Kindly define.
Line 286. ''Th experiment was...''.
Line 291 and elsewhere. ''66 kg''.
Line 296. ''Group A was...''.
Line 314.''..was calculated...''.
Line 316.''Hardness was measured...''.
Figure 9. Upper- or lowercase letters? Kindly be consistent in the Figure legends.
Lines 393, 398, 402, and elsewhere. Correct the reference format to be in agreement with the Guide to authors.
Line 426 and elsewhere.''From Table 7....''.
Line 489. ''In total, 7000 ...''.
Based on the aforementioned, I suggest a minor revision of the present well-built article.
Author Response

(The authors gave the same response as above.)

Round 2
Reviewer 2 Report
Dear Editor
Yes, it has been modified according to our suggestions.
I recommend the paper for publication.
Kindest regards,
Reviewer 3 Report
Now that I have read the other reviewers comments too, which I agree with, I come to the conclusion that this is only a borderline research paper (as far as the overall content goes). This would make a great (short) book chapter to show the application of the sensors in case of cherries.
This comment is more to point out to the authors that in the future a manuscript like this one doesn't really belong in this journal